# Determinants of maternity services utilisation among women of reproductive age across sub-Saharan Africa

Julia Marie Hajjar[1], Obasanjo Bolarinwa[2,3*], Oluwatobi Abel Alawode[4], Adeolu Anthony Olagunju[5], Lawrence Jones-Esan[2], Sanni Yaya[6,7]

1 Interdisciplinary School of Health Sciences, Faculty of Health Sciences, University of Ottawa, Ottawa, Ontario, Canada, 2 Department of Global Healthcare Management, York St John University, London, United Kingdom, 3 Department of Demography and Population Studies, University of Witwatersrand, Johannesburg, South Africa, 4 Department of Sociology and Criminology & Law, University of Florida, Gainesville, Florida, United States of America, 5 Department of Public Health & Wellbeing, Faculty of Medicine, Health & Society, University of Chester, Chester, United Kingdom, 6 School of International Development and Global Studies, Faculty of Social Sciences, University of Ottawa, Ottawa, Ontario, Canada, 7 The George Institute for Global Health, Imperial College London, London, United Kingdom

* bolarinwaobasanjo@gmail.com

## Abstract

### Background

In 2020, approximately 800 women died daily as a result of largely preventable complications of pregnancy and delivery globally. Almost 95% of these deaths occurred in low- and middle-income countries. Even though antenatal care, institutional delivery, and postnatal care constitute lifesaving maternal and newborn healthcare services, uptake is variable between countries in sub-Saharan Africa. Thus, this study examined the coverage and factors influencing the utilisation of maternal and newborn health services in sub-Saharan Africa.

### Methods

This current study pooled datasets from the Demographic Health Surveys conducted in 27 countries in sub-Saharan Africa between the years 2010–2020. The outcome variables were maternal and newborn health services measured by antenatal care visits, institutional delivery, and postnatal care visits among 58,648 women of reproductive age between the ages of 15–49. Multilevel analysis was employed to examine the associated factors at a $p < 0.05$ level of significance.

### Results

The overall analysis of the prevalence of maternal and newborn health services among women of reproductive age in sub-Saharan Africa was 67.3%, 74.5%, and 32.5% for 4 + antenatal care visits, institutional delivery and visits to postnatal care

**Data availability statement:** Data for this study is publicly available on the Demographic and Health Surveys (DHS) website (http://dhsprogram.com/data/available-datasets.cfm). Potential users must obtain written permission from ICF International before using the dataset. However, the merged dataset used in this study can be accessed through the following public data repository link https://osf.io/68fbr?view_only=e860300a4c0a4411ac2cdea471be3148.

**Funding:** The author(s) received no specific funding for this work.

**Competing interests:** The authors have declared that no competing interests exist.

**Abbreviations:** LMICs, low-middle-income countries; SSA, Sub-Saharan Africa; DHS, Demographic Health Survey; CI, Confidence Interval; ANC, Antenatal Care; PNC, Postnatal Care; SDG, Sustainable Development Goal; LLR, Log-likelihood Ratio; AIC, Akaike Information Criterion; BIC, Bayesian Information Criterion; VIF, Variance Inflation Factor; SES, Socioeconomic status; UHC, Universal Health Coverage.

within 48 hours of delivery, respectively. Antenatal care visits were highest in Sierra Leone at 91.4%, and Institutional delivery was highest in Gabon at 97.6%, whilst Niger had the lowest prevalence for antenatal care visits and institutional delivery at 38.0% and 42.3%, respectively. Cote d'Ivoire reported the highest prevalence of postnatal care with 78.8%, whilst Malawi reported the lowest with 7.3%. Moreover, women with secondary/higher education were more likely to utilise antenatal care (aOR=2.09; 95% CI:1.96–2.23) and have institutional delivery (aOR=2.54; 95% CI:2.34–2.74) compared to those with no education. Furthermore, being employed was associated with a higher likelihood of utilising postnatal care (aOR=1.28; 95% CI:1.22–1.34) within 48 hours of delivery compared to women without formal employment.

## Conclusion

The study concluded that women of reproductive age in sub-Saharan Africa who were educated were more likely to seek antenatal care and have institutional delivery, whilst women who were employed were more likely to utilise postnatal care within 48 hours of delivery. Therefore, future initiatives should focus on empowering and strengthening the education of girls and women in sub-Saharan Africa.

## Background

Despite the improvements in maternal health worldwide, women continue to die daily due to pregnancy and childbirth complications [1]. In 2020, approximately 800 women died daily as a result of largely preventable complications of pregnancy and delivery globally [1]. Almost 95% of these deaths occurred in low- and middle-income countries (LMICs) [2], with 70% occurring in sub-Saharan Africa (SSA). Women in SSA have the highest lifetime risk of maternal mortality [3] and account for 66% of the global maternal mortality ratio [4]. Even though antenatal care (ANC), institutional delivery, and postnatal care (PNC) constitute lifesaving maternal and newborn health-care services, uptake is variable between countries within SSA [5], and maternal mortality rates remain significantly higher than in the rest of the world [5]. Identifying facilitators and barriers to maternal and newborn health service coverage and quality is essential to improve health equity and reduce maternal and newborn morbidity and mortality rates in the region [4,5].

ANC is an essential maternal health service that provides preventative monitoring, education, and support throughout pregnancy [4,6]. ANC visits are a key strategy to reduce neonatal and maternal morbidity and mortality [7,8] as they provide the opportunity for a skilled healthcare professional to counsel pregnant women [9]. To reduce perinatal mortality, the 2016 World Health Organisation (WHO) ANC model guidelines increased the original recommendation of 4+ANC visits established in the 1990s to 8 ANC visits [10]. Despite its importance, global uptake of ANC services remains low [9,11]. Notably, in SSA, uptake of ANC services varies largely between countries [4], and just under half (49%) of women in SSA seek 4+ANC visits [12,13].

Institutional delivery is another important strategy to reduce maternal morbidity and mortality, particularly in SSA, as this ensures women receive care from qualified and skilled medical professionals during birth and postpartum [14,15]. However, almost half of the women in SSA do not receive appropriate care during delivery [16], with accessibility being a major deterrent to accessing health facilities [17,18]. PNC visits are one of the most important strategies to reduce maternal and newborn morbidity and mortality. PNC is defined as a mother and newborn receiving at least 1 PNC visit within 48 hours of childbirth and includes all women, regardless of place of delivery.. Therefore, it is vital to continue strengthening existing programs and services to ensure that women in SSA experience equitable access to life-saving maternal and newborn healthcare services, including ANC visits, access to healthcare facilities for delivery, and PNC visits. This will promote the upstream mitigation of largely preventable complications of pregnancy and childbirth that continue to take the lives of women and children in SSA.

To strengthen the coverage and quality of maternal and newborn health services in SSA, this study examined the factors associated with receiving i) 4+ANC visits, ii) institutional delivery and iii) receiving at least one PNC visit (mother and newborn receive at least 1 PNC visit within 48 hours of childbirth). Implementing strategies that are responsive to the unique geographic, economic, and sociocultural contexts throughout SSA will help support the progression towards achieving Sustainable Development Goal (SDG) target 3.1, which requires a reduction in maternal mortality rates under 70 per 100,000 live births globally by 2030 [19]. Thus, this study aims to contribute to the literature to advance knowledge surrounding the coverage and factors influencing maternal and newborn health services utilisation in SSA to reduce maternal and newborn morbidity and mortality in the region.

## Methods

### Study design and data source

This cross-sectional study employed Demographic Health Surveys (DHS) conducted in 27 countries in SSA. The survey that employed questionnaires was conducted in each country between the years 2010–2020. The eligible countries included in this study were countries with available datasets that measured outcomes and explanatory variables of interest. DHS is known for its nationally representative dataset, which includes data on various public health-related issues on maternal and newborn health, including ANC, institutional delivery & PNC in LMICs. [20]. DHS sampling procedure and data collection methods are outlined here [21]. A total sample size of 58,648 women of reproductive age between the ages of 15 and 49 was pooled from 27 countries in SSA between 2010 and 2020 and analysed. The included countries have DHS datasets that measured the outcome variables (ANC visit, institutional delivery, and PNC visit) and explanatory variables of interest, such as age, educational attainment, marital status, etc. The DHS datasets are available in the public domain and can be accessed via http://dhsprogram.com/data/available-datasets.cfm. Table 1 shows detailed information on the countries included in this study.

### Outcome variables

The outcome variables of interest in this study were maternal and newborn health services measured by ANC visit, institutional delivery, and PNC visit. For ANC visits, women of reproductive age who attended more than four (4+) ANC during the last pregnancy as recommended by WHO [10] were coded "yes"; otherwise were coded "no". For institutional delivery, those who delivered their last birth in any institutional delivery were coded "yes"; otherwise were coded "no". Lastly, those women of reproductive age who attended PNC within 48 hours of their last birth were coded "yes"; otherwise were coded "no". Similar measurements have been used in previous studies [22–24].

### Explanatory variables

The explanatory variables included in the study were age- defined as the age of the women of reproductive age in the study coded as 15–24 (1), 25–34 (2), and 35+ (3); the highest level of education for the women and their partners refers

**Table 1. Distribution of included countries by survey year, weighted sample, and percentage.**

| Country | Survey Year | Frequency | Percent |
|---------|-------------|-----------|---------|
| Angola | 2015/16 | 1,322 | 2.89 |
| Benin | 2017/18 | 1,428 | 3.13 |
| Burkina Faso | 2010 | 1,714 | 3.75 |
| Burundi | 2016/17 | 546 | 1.19 |
| CIV | 2011/12 | 967 | 2.12 |
| Cameroon | 2018/19 | 1,302 | 2.85 |
| Chad | 2014/15 | 512 | 1.12 |
| Congo | 2011/12 | 2,272 | 4.97 |
| CongoDR | 2013/14 | 1,607 | 3.52 |
| Ethiopia | 2016 | 555 | 1.21 |
| Gabon | 2012 | 822 | 1.8 |
| Gambia | 2019/20 | 2,332 | 5.1 |
| Guinea | 2018 | 1,622 | 3.55 |
| Kenya | 2014 | 1,154 | 2.53 |
| Lesotho | 2014 | 637 | 1.39 |
| Liberia | 2019/20 | 574 | 1.26 |
| Malawi | 2015/16 | 4,738 | 10.37 |
| Mali | 2018 | 1,472 | 3.22 |
| Niger | 2012 | 4,540 | 9.94 |
| Nigeria | 2018 | 3,201 | 7.01 |
| Rwanda | 2019/20 | 894 | 1.96 |
| Sierra-Leone | 2019 | 2,272 | 4.97 |
| South Africa | 2016 | 438 | 0.96 |
| Tanzania | 2015/16 | 743 | 1.63 |
| Togo | 2013 | 3,025 | 6.62 |
| Uganda | 2016 | 1,652 | 3.62 |
| Zimbabwe | 2015 | 3,340 | 7.31 |
| Total | | 45,681 | |

to the highest education that both the women and their partners in the survey have attained which was coded as none (1), primary (2) and secondary/higher (3). The study also included the working status of the women, i.e., whether they were employed at the time of the survey or not, which was coded as No(1) and Yes(1); the study was focused on married women and their type of marriage was coded as monogamy (1) and polygamy (2). For media exposure, the survey asked the respondents questions about the frequency of their exposure to mainstream media, including radio, television, and newspapers/magazines. Based on this, we developed a single variable that encompassed their responses and catego-rised their exposures into No (1) and Yes (1). Parity in this study is defined as the number of children born to women, which was a count variable, but for this study, we categorised them into 1–3 (1) and 4+ (2). Whether respondents wanted their last birth or not is an important variable in the survey that is associated with reproductive healthcare utilisation, and it was included in this study as a covariate coded as Wanted then (1), Wanted later (2), and Wanted no more (3); the sex of the child was also included and coded as male (1) and female (2); type of place of residence of the respondents was coded as urban 91) and rural (2); the survey also asked the respondents their perspective about the distance between their house and the nearest health facility and the responses were coded as a big problem (1) and not a big problem (2). Household and Community-level variables were also included in the study; the wealth index was developed based on

questions in the survey asking about the presence of several household properties, and this was used in this study as coded in the survey. The sex of the household head was coded as male (1) and female (2). Community-level variables were developed from individual-level variables, and a tercile was developed for both community-level literacy level and community socioeconomic status. All included variables were selected based on their influence on maternal and newborn health service utilisation in SSA [22–25].

## Statistical analyses

Datasets from the countries with variables of interest were pooled together using the STATA "append" command. Frequency analysis was performed as the first level of analysis in this study. Whilst chi-square analysis was used to test the association between the outcome and explanatory variables, then a three-level multi-level logistic regression was developed to assess the individual, household, and community level factors associated with ANC visit, institutional delivery, and PNC visit (maternal and newborn healthcare service utilisation). Women of reproductive age in SSA included in the study were nested within the household in the modelling, while the household/community were nested within the clusters. Clusters were the random effects that account for the unexplained variations at all levels. An empty "model 0" was first fitted for the random intercept interpretation, then "model I" to "model III" was fitted to count for individual level, household level, and community level variables, whilst the last "model IV" was fitted to account for all the explanatory variables. "melogit" stata command was used to fit the model which provided the odds ratio and its corresponding 95% confidence intervals (CIs). Model comparison was done using the log-likelihood ratio (LLR) and Akaike Information Criterion (AIC) to show which model significantly improves over others, whilst ensuring the appropriate model with goodness of fit is selected [26]. We also tested for multicollinearity of the included explanatory variables using the variance inflation factor (VIF), which showed no collinearity among the independent variables. In the individual populations, sample weight (v005/1,000,000) was used in all analyses to account for over-and undersampling. The "svy" command was used to account for the survey's complex nature, which also helps generalise the findings. Stata version 17.0 (Stata Corporation, College Station, TX, USA) was used for the statistical analyses.

## Ethics approval and consent to participate

Since the author of this manuscript did not collect the data, we sought permission from the MEASURE DHS website and access to the data was provided after our intent for request was assessed and approved on the 8th of January, 2023. The ORC Macro Inc. ethics committee and the ethics Boards of partner organisations in each SSA country, such as the Ministries of Health, ethically accept the DHS surveys. In line with the DHS informed consent guidelines, the interviewed women gave either written or verbal consent during the surveys in each country; additionally, parental consent was obtained from participants under 16 years old. All research methods adopted in this manuscript followed the relevant guidelines and regulations in line with the stipulations of the World Medical Association Declaration of Helsinki Ethical Principles [27].

## Results

The analysis revealed that the prevalence of maternal and newborn health services among women of reproductive age in SSA was 67.3%, 74.5%, and 32.5% for 4+ANC visits, institutional delivery, and visits for PNC within 48 hours of delivery, respectively. ANC visits of 4+ were highest in Sierra Leone at 91.4% and lowest in Niger, where 38.0% of the women had 4+ANC visits during pregnancy. It was found that the highest percentage of women who had institutional delivery can be found in Gabon at 97.6%, followed closely by South Africa at 96.9%, while Niger has the lowest percentage of women delivering at a health facility with 42.3%. The analysis further showed that the highest prevalence of women reporting postnatal care 48 hours after delivery can be found in Cote d'Ivoire (78.8%), and the lowest percentage can be found in Malawi at 7.3% (Fig 1).

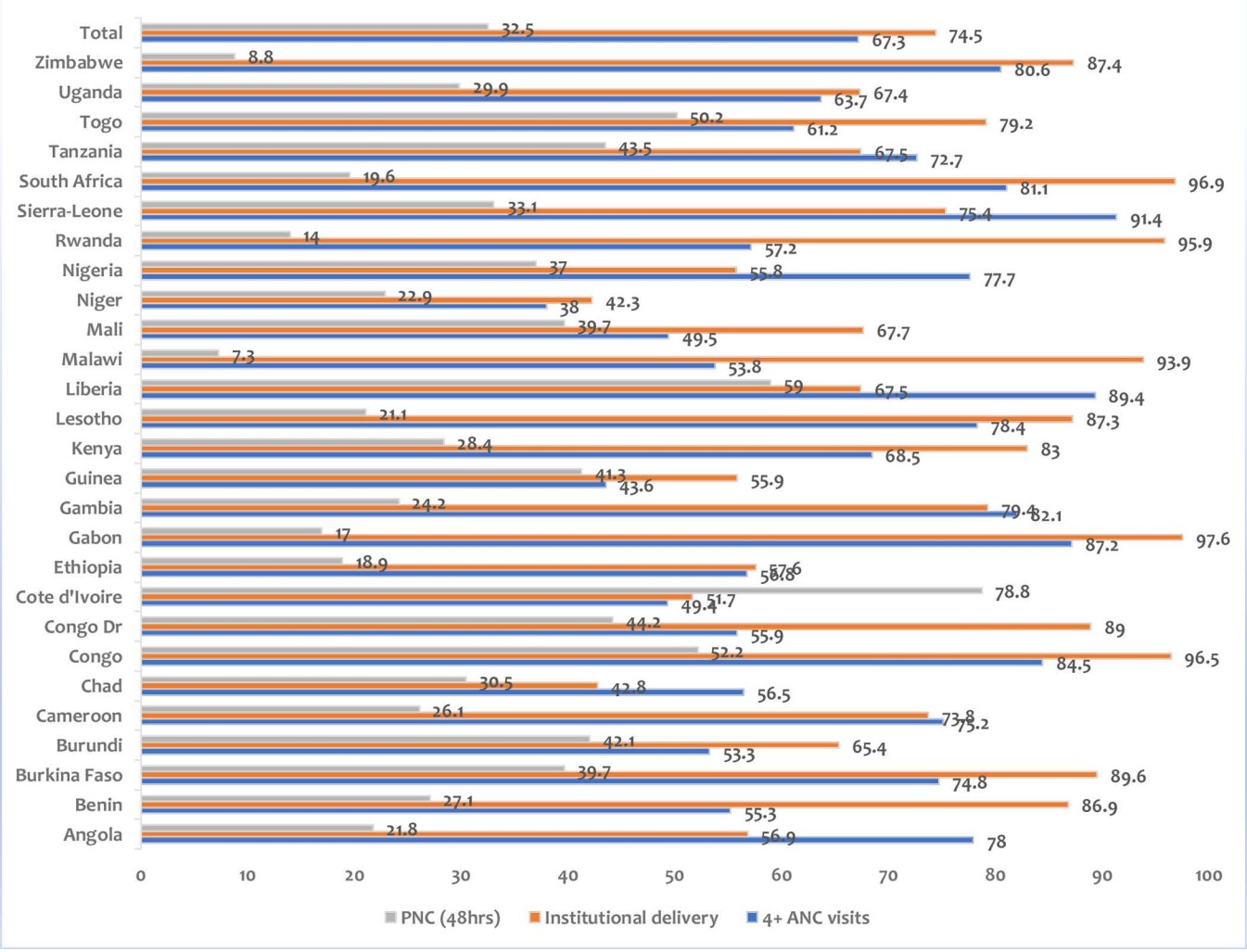

**Fig 1. Percentage distribution of Postnatal Care within 48 hours of Child Delivery, Institutional delivery, and 4 + antenatal visits among women of reproductive age in SSA.**

Table 2 shows the distribution of the respondents by individual, household, and community-level characteristics, as well as the bivariate distribution of maternal healthcare utilisation by the individual, household, and community-level characteristics. It can be reported that 49% of the women are between 25–34 years, while 26% are young adults. Also, 35% of the respondents have no formal education, while 30% reported having a secondary/higher level of education, and about 31% and 25% reported that their partners have no formal education and primary education, respectively. It was also found that 71% of the respondents reported working, and 72% were exposed to mass media. It was also found that 78% of the women are in monogamous marriages, while more than half of the respondents have had between 1–3 children (55%), and about 37% of the women reported residing in urban areas.

The analysis also showed that 76% of the women said they wanted their last birth when they had it, and 51% of the women had a male child at their last birth. The analysis showed that about 62% of the women reported that the distance from their home to the closest health facility is not a big problem. Approximately 20% of the women reported belonging to each of the household wealth categories: poorer, middle, richer, and richest. Regarding community-level factors, 34% of

**Table 2. The distribution of respondents and the bivariate table showing the distribution of maternal healthcare services by respondents' characteristics.**

| | Frequency | Percent | 4+ANC | Institutional Delivery | 48 hours PNC Check |
|---|---|---|---|---|---|
| **Age groups** | | | <0.001 | <0.001 | <0.001 |
| 15-24 | 11,939 | 26.1 | 61.6 | 74.7 | 27.6 |
| 25-34 | 22,210 | 48.6 | 66.5 | 74.2 | 28.7 |
| 35-49 | 11,527 | 25.2 | 64.4 | 70.5 | 30.9 |
| **Maternal Level of Education** | | | <0.001 | <0.001 | <0.001 |
| None | 15,983 | 34.9 | 53.4 | 56.3 | 35.7 |
| Primary | 13,601 | 29.8 | 62.2 | 77 | 26.3 |
| Sec/Higher | 16,093 | 35.2 | 79.1 | 88.8 | 24.1 |
| **Partner Level of Education** | | | <0.001 | <0.001 | <0.001 |
| None | 14,132 | 30.9 | 53.7 | 56.6 | 34.9 |
| Primary | 11,292 | 24.7 | 60.3 | 74.7 | 26.3 |
| Sec/Higher | 20,253 | 44.3 | 75.6 | 85.5 | 26 |
| **Work Status** | | | <0.001 | <0.001 | <0.001 |
| No | 13,184 | 28.9 | 59.7 | 69.8 | 25.2 |
| Yes | 32,493 | 71.1 | 66.7 | 74.9 | 30.5 |
| **Type of marriage** | | | <0.001 | <0.001 | <0.001 |
| Monogamy | 35,688 | 78.1 | 66.3 | 76.7 | 27.7 |
| Polygamy | 9,989 | 21.9 | 58.9 | 62.1 | 33.2 |
| **Mass media exposure** | | | <0.001 | <0.001 | <0.001 |
| No | 12,938 | 28.3 | 55.9 | 63.5 | 32.7 |
| Yes | 32,739 | 71.7 | 68.1 | 77.4 | 27.5 |
| **Parity** | | | <0.001 | <0.001 | <0.001 |
| 1–3 | 25,301 | 55.4 | 68.4 | 80 | 26.9 |
| 4+ | 20,376 | 44.6 | 60 | 65.3 | 31.6 |
| **Type of place of residence** | | | <0.001 | <0.001 | <0.001 |
| Urban | 16,834 | 36.9 | 75.9 | 86.9 | 28.7 |
| Rural | 28,843 | 63.2 | 58.5 | 66 | 29.1 |
| **Wanted last-child** | | | <0.001 | <0.001 | 0.001 |
| Wanted Then | 34,636 | 75.8 | 65.5 | 72.1 | 29.5 |
| Wanted Later | 8,489 | 18.6 | 62.2 | 77.2 | 27.8 |
| Wanted No More | 2,552 | 5.6 | 61.8 | 78.3 | 25.6 |
| **Sex of last birth/child** | | | <0.001 | 0.0091 | <0.001 |
| Male | 23,120 | 50.6 | 64.4 | 74.1 | 29.1 |
| Female | 22,557 | 49.4 | 64.9 | 72.7 | 28.8 |
| **Distance to Health Facility** | | | <0.001 | <0.001 | <0.001 |
| Big Problem | 17,382 | 38.1 | 58.6 | 65.4 | 29.1 |
| Not a big problem | 28,295 | 61.9 | 68.4 | 78.3 | 28.9 |
| **Household Wealth Index** | | | <0.001 | <0.001 | <0.001 |
| Poorest | 8,560 | 18.7 | 55.3 | 56.8 | 33.3 |
| Poorer | 9,108 | 19.9 | 60.3 | 66.4 | 30.3 |
| Middle | 9,069 | 19.9 | 61.9 | 70.2 | 29.8 |
| Richer | 9,276 | 20.3 | 67.2 | 80.2 | 26.4 |
| Richest | 9,664 | 21.2 | 77.3 | 91.2 | 26.1 |
| **Sex of Household Head** | | | <0.001 | <0.001 | <0.001 |
| Male | 38,815 | 84.9 | 64.2 | 72.6 | 29.5 |

*(Continued)*

**Table 2.** (Continued)

| | Frequency | Percent | 4+ANC | Institutional Delivery | 48 hours PNC Check |
|---|---|---|---|---|---|
| Female | 6,862 | 15.1 | 67.4 | 77.8 | 26.2 |
| **Community Literacy Level** | | | **<0.001** | **<0.001** | **<0.001** |
| Low | 14,301 | 31.3 | 61.7 | 68.5 | 31.1 |
| Medium | 14,972 | 32.8 | 64.2 | 72.2 | 30.2 |
| High | 16,404 | 35.9 | 67.7 | 78.9 | 25.9 |
| **Community Socioeconomic Status** | | | **<0.001** | **<0.001** | **<0.001** |
| Low | 14,440 | 31.6 | 62.4 | 68.9 | 29.2 |
| Medium | 15,298 | 33.5 | 63.9 | 72.9 | 30.3 |
| High | 15,938 | 34.9 | 67.4 | 77.9 | 28.9 |

the women are from communities with high socioeconomic status, while about 35% are from communities characterised by both high socioeconomic status and high literacy rates.

Furthermore, the results of the χ2 indicated that all the individual, community, and household level variables had a significant association with antenatal care utilisation, aside from fertility desire, sex of the last child, and community socioeconomic status. For institutional delivery, it was found that all the individual, household, and community-level variables are associated with delivering at a health facility, aside from community socioeconomic status. Lastly, age, exposure to mass media, sex of last child, household wealth index, sex of household head, and community-level variables were not significantly associated with postnatal care 48 hours after the last child's birth.

Table 3 shows the results of the multilevel regression analysis of the factors associated with antenatal care utilisation among the sample of women. The complete model (Model IV) presents the full and most viable model of ANC-associated factors based on the associated model diagnostics, including the Akaike Information Criterion. The results indicate that older women were significantly more likely to have had 4+ANC visits during pregnancy; women aged 35−49 were 47% more likely to have had 4+ANC visits during pregnancy compared to women aged 15−24 (aOR=1.47; 95% CI:1.37–1.58). Secondary/higher education was found to be significantly associated with 2 times higher likelihood of 4+ANC visits during pregnancy compared to having no formal education (aOR=2.09; 95% CI: 1.96–2.23), and this also applies to women who reported partners with secondary/higher education as they were 45% more likely to have 4+ANC visits during pregnancy (aOR=1.45; 95% CI: 1.37–1.54) compared to women with no education and those whose partner had no educational level. Women who were employed at the time of the survey were 30% more likely to have 4+ANC during pregnancy compared to those who were not working (aOR=1.30; 95% CI: 1.25–1.36).

Furthermore, women in polygamous marriages were 5% less likely to have 4+ANC visits compared to women in monogamy (aOR= 0.95; 95% CI: 0.90–0.99), high parity women (4+children) were also found to be 18% significantly less likely to have 4+ANC visits during pregnancy compared to women with 1−3 children (aOR= 0.82; 95% CI: 0.78–0.86). Women who reported wanting their last birth later (aOR=0.76; 95% CI: 0.72–0.80) and did not want any more (aOR=0.76; 95% CI: 0.69–0.86) were both 24% less likely to have 4+ANC visits during their last pregnancy. Women who stated that the distance from their home to a health facility "is not a big problem" were 18% significantly more likely to have 4+ANC (aOR=1.18; 95% CI: 1.13–1.24) compared to those who stated that distance to a health facility "is a big problem", while it was found that rural dwellers are 28% significantly less likely to have 4+ANC visits (aOR=0.72; 95% CI: 0.68–0.76) compared to urban dwellers. The result also showed an inverse relationship between the household wealth index and ANC visits.

The comparison model shows that community-level factors play a significant role in ANC utilisation, with the ICC increasing from 2.5% in the null model to 9% in the fully adjusted model (Model IV). The likelihood ratio tests are

**Table 3.  Results of multilevel logistic regression analysis of the determinants of antenatal care utilisation among women in sub-Saharan Africa.**

| Antenatal Care | Model 0 | Model I [Individual Level] aOR | Model II [Household Level] aOR | Model III [Community Level] aOR | Model IV [All Variables] aOR |
|---|---|---|---|---|---|
| **Fixed Effect** | | | | | |
| Maternal age [ref: 15–24] | | | | | |
| 25-34 | | 1.32 *** [1.25 - 1.39] | | | 1.29 *** [1.22 - 1.37] |
| 35-49 | | 1.51 *** [1.40 - 1.62] | | | 1.47 *** [1.37 - 1.58] |
| **Maternal education [ref: None]** | | | | | |
| Primary | | 1.28 *** [1.21 - 1.36] | | | 1.28 *** [1.21 - 1.36] |
| Sec/Higher | | 1.23 *** [1.01 - 1.51] | | | 2.09 *** [1.96 - 2.23] |
| **Paternal Education [ref: None]** | | | | | |
| Primary | | 1.03 [0.97 - 1.09] | | | 1.04 [0.98 - 1.10] |
| Sec/Higher | | 1.49 *** [1.41 - 1.59] | | | 1.45 *** [1.37 - 1.54] |
| **Work Status [ref: No]** | | | | | |
| Yes | | 1.29 *** [1.23 - 1.35] | | | 1.30 *** [1.25 - 1.36] |
| **Type of Marriage [ref: Monogamy]** | | | | | |
| Polygamy | | 0.94 ** [0.90 - 0.99] | | | 0.95 ** [0.90 - 0.99] |
| **Mass Media Exposure [ref: No]** | | | | | |
| Yes | | 1.23 *** [1.17 - 1.28] | | | 1.19 *** [1.14 - 1.25] |
| **Parity [ref: 1–3]** | | | | | |
| 4+ | | 0.80 *** [0.76 - 0.85] | | | 0.82 *** [0.78 - 0.86] |
| **Fertility desire [ref: Wanted then]** | | | | | |
| Wanted later | | 0.76 *** [0.72 - 0.80] | | | 0.76 *** [0.72 - 0.80] |
| Wanted no more | | 0.76 *** [0.70 - 0.83] | | | 0.76 *** [0.69 - 0.83] |
| **Sex of Child [ref: Male]** | | | | | |
| Female | | 0.99 [0.95 - 1.03] | | | 0.99 [0.95 - 1.03] |
| **Perception of Distance to Nearest Health Facility [ref: A big problem]** | | | | | |
| Not a big problem | | 1.23 *** [1.18 - 1.28] | | | 1.18 *** [1.13 - 1.24] |
| **Type of Place of Residence [ref: Urban]** | | | | | |
| Rural | | | 0.59 *** [0.56 - 0.63] | | 0.72 *** [0.68 - 0.76] |
| **Household Wealth Index [ref: Poorest]** | | | | | |
| Poorer | | | 1.17 [1.10 - 1.24] | | 1.06 [0.99 - 1.12] |
| Middle | | | 1.15 *** [1.08 - 1.22] | | 0.97 [0.91 - 1.03] |
| Richer | | | 1.21 *** [1.13 - 1.29] | | 0.91 ** [0.85 - 0.98] |
| Richest | | | 1.50 *** [1.39 - 1.61] | | 0.94 [0.87 - 1.02] |
| **Sex of Household Head [ref: Male]** | | | | | |
| Female | | | 1.19 *** [1.12 - 1.26] | | 1.06 ** [1.01 - 1.13] |
| **Community Literacy Level [ref: Low]** | | | | | |
| Medium | | | | 1.09 ** [1.01 - 1.17] | 1.02 [0.95 - 1.10] |
| High | | | | 1.18 *** [1.10 −1.27] | 0.95 [0.88 - 1.02] |
| **Community Socioeconomic Status [ref: Low]** | | | | | |
| Medium | | | | 1.01 [0.94 - 1.09] | 0.99 [0.95 - 1.10] |
| High | | | | 1.13 ** [1.05 - 1.22] | 0.95 [0.88 - 1.05] |

*(Continued)*

**Table 3.** (Continued)

| | Model 0 | Model I [Individual Level] | Model II [Household Level] | Model III [Community Level] | Model IV [All Variables] |
|---|---|---|---|---|---|
| **Random Effect** | | | | | |
| PSU Variance [95% CI] | 0.7 [0.6 - 0.8] | 0.07 [0.05 - 0.09] | 0.08 [0.06 - 0.10] | 0.08 [0.06 - 0.10] | 0.07 [0.05 - 0.09] |
| ICC | 2.50% | 2.00% | 2% | 2.00% | 9% |
| Likelihood Ratio Test | χ2 = 240.07; p < 0.001 | χ2 = 183.23; p < 0.001 | χ2 = 239.87; p < 0.001 | χ2 = 215.40; p < 0.001 | χ2 = 188.70; p < 0.001 |
| Walds | Reference | 2764.31 | 1015 | 52.69 | 2884.26 |
| Log-Likelihood | −29720.42 | −28194.82 | −29186.4 | −29693.84 | −28115.24 |
| AIC | 59,444.83 | 56421.65 | 58388.79 | 59399.68 | 56282.48 |

CI: Confidence Interval; aOR: adjusted Odds ratio; ***p < 0.001, ** p < 0.01.

significant across all models (p < 0.001), indicating that individual, household, and community factors contribute to ANC usage.

Table 4 shows the multilevel regression result of the factors associated with institutional (health facility) delivery among young women in SSA. In the full model (Model IV), which is also the most viable based on the estimates from the random effects, it can be reported that the older a woman, the more likely she was to deliver at a health facility with women aged 25−34 and 35−49 being 19% (aOR=1.18; 95% CI:1.10–1.26) and 39% (aOR=1.39; 95% CI: 1.28–1.51) significantly more likely to deliver at a health facility respectively. For education, women who report secondary/higher education were found to be more than 2 times more likely to deliver at a health facility compared to women with no formal education (aOR=2.54; 95% CI: 2.34–2.74). Women who reported that their spouses have higher education had a 68% higher likelihood of delivering at a health facility compared to women with partners with no formal education (aOR= 1.68; 95% CI: 1.56–1.80). The study also found that women with 4 + children were 37% significantly less likely to have had a birth at a health facility compared to women with 1−3 children (aOR=0.63; 95% CI: 0.59–0.67) and women who reported that they didn't want any more children were found to be 25% significantly more likely to deliver at a health facility compared to women who wanted the child then (aOR=1.25; 95% CI: 1.12–1.40).

Furthermore, the analysis showed that women who stated that the distance of their home to the nearest health facility is not a problem were 38% significantly more likely to deliver at a health facility than those who reported that it is a big problem (aOR=1.38; 95% CI:1.31–1.45). Women residing in rural areas were found to be 34% less likely to deliver at a health facility compared to women in urban areas (aOR=0.66; 95% CI: 0.61–0.71). For other household-level variables, the analysis further revealed that a higher household wealth index is associated with a higher likelihood of delivering at a health facility. Women from female-headed households were found to be 13% more likely to deliver at a health facility compared to women from male-headed households (aOR=1.13; 95% CI:1.06–1.21). It can be reported that women residing in medium literacy level communities were 17% more likely to deliver at a health facility compared to women from communities with low literacy (aOR=1.17; 95% CI: 1.03–1.31) while women residing in medium level socioeconomic status (SES) communities were 14% more likely to deliver at a health facility compared to women from communities with low SES communities (aOR=1.14; 95% CI: 1.02–1.27).

The comparison model shows that the ICC reaches 8% in the fully adjusted model, showing a strong community influence on facility-based births. The model fit improves significantly, with the lowest AIC at 44,891.64, confirming that community-level factors are critical to understanding facility delivery utilisation in SSA.

Table 5 presents the results of the multilevel regression analysis of the factors associated with PNC 48 hours after delivery among the sample of women. The result showed that women with primary (aOR=0.93; 95% CI:0.87–0.98) and

**Table 4. Results of multilevel logistic regression analysis of the determinants of Health Facility Delivery among women in sub-Saharan Africa.**

| | Model 0 | Model I [Individual Level] | Model II [Houeshold Level] | Model III [Community Level] | Model IV [All Variables] |
|---|---|---|---|---|---|
| Institutional Delivery | | aOR | aOR | aOR | aOR |
| **Fixed Effect** | | | | | |
| **Maternal age [ref: 15–24]** | | | | | |
| 25-34 | | 1.29 *** [1.21 - 1.38] | | | 1.18 *** [1.10 - 1.26] |
| 35-49 | | 1.56 *** [1.43 - 1.70] | | | 1.39 *** [1.28 - 1.51] |
| **Maternal education [ref: None]** | | | | | |
| Primary | | 1.88 *** [1.77 - 2.00] | | | 1.89 *** [1.78 - 2.02] |
| Sec/Higher | | 2.99 *** [2.77 - 3.23] | | | 2.54 *** [2.34 - 2.74] |
| **Paternal Education [ref: None]** | | | | | |
| Primary | | 1.47 *** [1.37 - 1.57] | | | 1.46 *** [1.37 - 1.56] |
| Sec/Higher | | 1.94 *** [1.81 - 2.08] | | | 1.68 *** [1.56 - 1.80] |
| **Work Status [ref: No]** | | | | | |
| Yes | | 1.15 *** [1.10 - 1.22] | | | 1.27 *** [1.20 - 1.34] |
| **Type of Marriage [ref: Monogamy]** | | | | | |
| Polygamy | | 0.78 *** [0.74 - 0.82] | | | 0.78 *** [0.74 - 0.82] |
| **Mass Media Exposure [ref: No]** | | | | | |
| Yes | | 1.26 *** [1.20 −.132] | | | 1.04 [0.99 - 1.10] |
| **Parity [ref: 1–3]** | | | | | |
| 4+ | | 0.58 *** [0.54 - 0.61] | | | 0.63 *** [0.59 - 0.67] |
| **Fertility desire [ref: Wanted then]** | | | | | |
| Wanted later | | 0.99 [0.94 - 1.06] | | | 1.04 [0.97 - 1.01] |
| Wanted no more | | 1.25 *** [1.13 - 1.40] | | | 1.25 *** [1.12 - 1.40] |
| **Sex of Child [ref: Male]** | | | | | |
| Female | | 0.93 ** [0.88 - 0.97] | | | 0.92 ** [0.88 - 0.97] |
| **Perception of Distance to Nearest Health Facility [ref: A big problem]** | | | | | |
| Not a big problem | | 1.63 *** [1.55 - 1.70] | | | 1.38 *** [1.31 - 1.45] |
| **Type of Place of Residence [ref: Urban]** | | | | | |
| Rural | | | 0.53 *** [0.49 - 0.56] | | 0.66 *** [0.61 - 0.71] |
| **Household Wealth Index [ref: Poorest]** | | | | | |
| Poorer | | | 1.47 *** [1.38 - 1.56] | | 1.28 *** [1.20 - 1.37] |
| Middle | | | 1.62 *** [1.52 - 1.73] | | 1.33 *** [1.24 - 1.42] |
| Richer | | | 2.39 *** [2.22 - 2.57] | | 1.77 *** [1.63 - 1.91] |
| Richest | | | 5.24 *** [4.75 - 5.79] | | 3.30 *** [2.97 - 3.68] |
| **Sex of Household Head [ref: Male]** | | | | | |
| Female | | | 1.33 *** [1.25 - 1.42] | | 1.13 *** [1.06 - 1.21] |
| **Community Literacy Level [ref: Low]** | | | | | |
| Medium | | | | 1.26 *** [1.14 - 1.40] | 1.17 ** [1.03 - 1.31] |
| High | | | | 1.64 *** [1.49 - 1.81] | 1.07 [0.95 - 1.20] |

*(Continued)*

**Table 4.** (Continued)

| | Model 0 | Model I [Individual Level] | Model II [Houeshold Level] | Model III [Community Level] | Model IV [All Variables] |
|---|---|---|---|---|---|
| **Community Socioeconomic Status [ref: Low]** | | | | | |
| Medium | | | | 1.13 ** [1.02 - 1.24] | 1.14 ** [1.02 - 1.27] |
| High | | | | 1.29 *** [1.17 - 1.42] | 0.95 [0.85 - 1.06] |
| **Random Effect** | | | | | |
| PSU Variance [95% CI] | 0.26 [0.21 - 0.30] | 0.25 [0.21 - 0.31] | 0.24 [0.20 −.029] | 0.19 [0.16 - 0.23] | 0.27 [0.22 - 0.33] |
| ICC | 7.00% | 7.00% | 2% | 5.00% | 8% |
| Likelihood Ratio Test | χ2 = 813.21; p < 0.001 | χ2 = 597.58; p < 0.001 | χ2 = 677.67; p < 0.001 | χ2 = 580.32; p < 0.001 | χ2 = 602.82; p < 0.001 |
| Walds | Reference | 4867.14 | 2895.97 | 189.11 | 5333.28 |
| Log-Likelihood | −25929.72 | −23016.22 | −24121.02 | −25837.82 | −22419.82 |
| AIC | 51,863.44 | 46064.44 | 48258.04 | 51687.64 | 44891.64 |

CI: Confidence Interval; aOR: adjusted Odds ratio; ***$p < 0.001$, ** $p < 0.01$.

secondary/higher education (aOR=0.91; 95% CI: 0.85–0.98) were 7% and 9% significantly less likely to have postnatal care within 48 hours of delivery compared to women with no formal education, and the same result was obtained for partner education, with women whose partners have primary and higher education showing a lower likelihood of having PNC care 48 hours after delivery compared to women whose partners had no formal education. The analysis further showed that a woman being employed was associated with a 28% higher likelihood of having PNC within 48 hours of delivery than women with no formal employment (aOR=1.28; 95% CI: 1.22–1.34). Women in Polygamous marriages were found to be 10% more likely to have had postnatal care compared to monogamous women (aOR=1.09; 95% CI: 1.05–1.16).

It was also found that women with mass media exposure were found to be 11% less likely to have PNC within 48 hours of delivery compared to those with no mass media exposure (aOR=0.89; 95% CI: 0.85–0.94), and in addition, higher parity women are 13% more to have had PNC at last birth compared to lower parity women (aOR=1.14; 95% CI: 1.08–1.20). The analysis also revealed that women who did not want their last birth were 14% less likely to have a PNC within 48 hours of delivery compared to women who wanted it (aOR=0.86; 95% CI: 0.78–0.94). It was found that women from households with a higher wealth index are less likely to get postnatal care within 48 hours of delivery, which can be regarded as an inverse relationship. The analysis also showed that residing in rural areas is associated with a 20% lower likelihood of postnatal care after delivery compared to urban dwellers (aOR=0.80; 95% CI: 0.76–0.84). Women from female-headed households were found to be 16% less likely to get postnatal care within 48 hours of child delivery compared to women from male-headed households (aOR=0.84; 95% CI: 0.79–0.88). The result at the community level showed that women in high literacy communities are 19% less likely to have had postnatal checks within 48 hours of delivery (aOR=0.81; 95% CI: 0.74–0.88).

The model comparison results show that community-level factors also affect PNC utilisation, with a stable ICC of 4–5% across models, indicating moderate community influence. The fully adjusted model has the lowest AIC at 56,019.74, demonstrating that community, along with individual and household factors, significantly contributes to PNC uptake within 48 hours of delivery.

**Table 5. Results of multilevel logistic regression analysis of the determinants of Postnatal Care among women in sub-Saharan Africa.**

| | Model 0 | Model I [Individual Level] | Model II [Houeshold Level] | Model III [Community Level] | Model IV [All Variables] |
|---|---|---|---|---|---|
| Postnatal Care | | aOR | aOR | aOR | aOR |
| **Fixed Effect** | | | | | |
| **Maternal age [ref: 15–24]** | | | | | |
| 25-34 | | 0.91 *** [0.86 - 0.96] | | | 0.93 ** [0.87 - 0.98] |
| 35-49 | | 0.89 *** [0.83 - 0.95] | | | 0.91 ** [0.85 - 0.98] |
| **Maternal education [ref: None]** | | | | | |
| Primary | | 0.76 *** [0.72 - 0.80] | | | 0.77 *** [0.73 - 0.82] |
| Sec/Higher | | 0.68 *** [0.64 - 0.73] | | | 0.71 *** [0.67 - 0.76] |
| **Paternal Education [ref: None]** | | | | | |
| Primary | | 0.84 *** [0.79 - 0.89] | | | 0.86 *** [0.81 - 0.91] |
| Sec/Higher | | 0.96 [0.90 - 1.02] | | | 1.01 [0.94 - 1.07] |
| **Work Status [ref: No]** | | | | | |
| Yes | | 1.30 *** [1.24 - 1.36] | | | 1.28 *** [1.22 - 1.34] |
| **Type of Marriage [ref: Monogamy]** | | | | | |
| Polygamy | | 1.09 ** [1.04 - 1.14] | | | 1.10 *** [1.05 - 1.16] |
| **Mass Media Exposure [ref: No]** | | | | | |
| Yes | | 0.85 *** [0.82 - 0.89] | | | 0.89 *** [0.85 - 0.94] |
| **Parity [ref: 1–3]** | | | | | |
| 4+ | | 1.17 *** [1.11 - 1.23] | | | 1.14 *** [1.08 - 1.20] |
| **Fertility desire [ref: Wanted then]** | | | | | |
| Wanted later | | 1.04 [0.98 - 1.10] | | | 1.03 [0.98 - 1.09] |
| Wanted no more | | 0.85 ** [0.77 - 0.94] | | | 0.86 ** [0.78 - 0.94] |
| **Sex of Child [ref: Male]** | | | | | |
| Female | | 0.98 [0.94 - 1.02] | | | 0.98 [0.94 - 1.02] |
| **Distance2HF [ref: A big problem]** | | | | | |
| Not a big problem | | 1.07 ** [1.02 - 1.12] | | | 1.10 *** [1.05 - 1.15] |
| **Type of Place of residence [ref: Urban]** | | | | | |
| Rural | | | 0.84 *** [0.79 - 0.89] | | 0.80 *** [0.76 - 0.84] |
| **Household Wealth Index [ref: Poorest]** | | | | | |
| Poorer | | | 0.84 *** [0.79 - 0.89] | | 0.87 *** [0.82 - 0.92] |
| Middle | | | 0.73 *** [0.69 - 0.78] | | 0.77 *** [0.73 - 0.83] |
| Richer | | | 0.57 *** [0.53 - 0.61] | | 0.63 *** [0.59 - 0.68] |
| Richest | | | 0.54 *** [0.49 - 0.58] | | 0.63 *** [0.58 - 0.68] |
| **Sex of Household Head [ref: Male]** | | | | | |
| Female | | | 0.81 *** [0.76 - 0.86] | | 0.84 *** [0.79 - 0.89] |
| **Community Literacy Level [ref: Low]** | | | | | |
| Medium | | | | 0.93 [0.85 - 1.02] | 0.98 [0.89 - 1.07] |
| High | | | | 0.71 *** [0.64 - 0.77] | 0.81 *** [0.74 - 0.88] |
| **Community Socioeconomic Status [ref: Low]** | | | | | |
| Medium | | | | 1.08 [0.99 - 1.18] | 1.07 [0.98 - 1.17] |
| High | | | | 1.02 [0.94 - 1.17] | 1.05 [0.96 - 1.15] |

*(Continued)*

**Table 5.** (Continued)

| | Model 0 | Model I [Individual Level] | Model II [Houeshold Level] | Model III [Community Level] | Model IV [All Variables] |
|---|---|---|---|---|---|
| **Random Effect** | | | | | |
| PSU Variance [95% CI] | 0.15 [0.13 - 0.19] | 0.14 [0.12 - 0.18] | 0.14 [0.12 - 0.18] | 0.19 [0.16 - 0.23] | 0.14 [0.11 - 0.17] |
| ICC | 5.00% | 4.00% | 4% | 4.00% | 4% |
| Likelihood Ratio Test | χ2 = 450.48; p < 0.001 | χ2 = 419.74; p < 0.001 | χ2 = 415.72; p < 0.001 | χ2 = 420.76; p < 0.001 | χ2 = 404.03; p < 0.001 |
| Walds | Reference | 678.55 | 406.05 | 82.22 | 924.53 |
| Log-Likelihood | −28458.25 | −28114.13 | −28252.99 | −28415.92 | −27983.87 |
| AIC | 56,920.51 | 56260.27 | 56521.89 | 56843.84 | 56019.74 |

CI: Confidence Interval; aOR: adjusted Odds ratio; ***$p < 0.001$, ** $p < 0.01$.

## Discussion

This current study examines the determinants of maternity services utilisation using a multi-level analysis across SSA. The results from the study show that ANC visits were highest for women in Sierra Leone at 91.5% and lowest in Niger, where only 38.0% of women received 4+ANC visits during pregnancy. Women were significantly more likely to have 4+ANC visits if they were between the ages of 25–49, if they (and their partners) had secondary education or higher, if they were employed and exposed to mass media, or if the distance to the health facility was not a big problem. Women were significantly less likely to have 4+ANC visits if they were in a polygamous marriage, had high parity, wanted a child later or wanted no more if they resided in a rural area, and if the woman was richer a wealth index. In SSA, approximately half of women receive 4+ANC visits during pregnancy [28]. Consistent with the findings of this study, previous studies have indicated that 4+ANC visits are more likely if women have secondary or higher education, if women are employed, and if women do not consider the distance to a health facility as a problem [28,29]. However, contrary to our findings, the literature suggests that low ANC uptake is associated with lower household income in LMICs [28,30], with poverty preventing women from accessing health services [28].

Similarly, in congruence with the findings of this study, previous investigations have identified that distance to a healthcare facility remains a barrier to accessing and using ANC services [4] and that women were more likely to have completed 4+ANC visits during pregnancy if they did not find the distance to the healthcare facility problematic. In SSA, women living in urban areas are also more likely to receive ANC than women in rural areas [31]. Studies on the impact of polygamous marriage vary, with some reporting no impact on 4+ANC visits [32], while others hypothesise that having a husband provides women with the financial means required to seek maternal health resources during pregnancy [33]. In Nigeria and Ghana, the ability to use health insurance has also been identified as a factor that impacts ANC visits [31,34,35]. Low socioeconomic status and living in rural areas may negatively impact the number of ANC visits [29].

In this study, the greatest percentage of women who delivered at a health facility was in Gabon (97.6%), closely followed by South Africa (96.9%), while Niger had the lowest rate at 42.3%. Women were significantly more likely to have a health facility delivery if they were between the ages of 25–49 and if they (and their partners) had primary or secondary education or higher, and if the distance to the health facility was not a big problem. Other factors associated with a higher likelihood of health facility delivery were if women were employed, wanted no more children, belonged to a medium SES community and were of a higher household wealth index. Factors associated with a lower likelihood of health facility delivery include having high parity (4+children), being in a polygamous marriage, if the sex of their last child was female and if they resided in a rural area. In many parts of SSA, less than 50% of births occur in health facilities [36]. Consistent with the findings from our study, a systematic review reported that women with high parity were less likely to give birth in

a health facility in SSA [37]. This was similarly observed in a Nigerian study [38], which suggests that this finding may be attributed to the fact that larger family sizes may result in financial barriers to accessing health facilities [38]. A previous study in Benin showed that women in high-literacy communities are less likely to give birth at home compared to health-care facilities [39]. In Eritrea, women who live a shorter distance from health facilities have also been shown to be more likely to give birth in a health facility [40]. In Ghana and Kenya, giving birth with skilled birth attendants present was more likely if women were wealthier, had higher education and lived in urban areas [41].

In this study, the highest rate of PNC visits occurred in Cote-d'Ivoire (78.8%), while only 7.3% of women received PNC in Malawi. Women were significantly more likely to receive at least one PNC visit if they were employed, in a polygamous marriage, had high parity, and if distance to a health facility was not a big problem. Women were significantly less likely to receive at least one PNC visit if they had primary, secondary, or higher education or if their partner had only primary education. Women were also significantly less likely to receive at least one PNC visit if they were exposed to mass media, if they did not want more children, resided in a rural area, had higher household wealth, had sex of the household head was female, or if their community's literacy level was high. In SSA, the rate of PNC is only 32.5%, while it is almost 86% in North Africa [42]. The literature indicates that pregnancy desire increases the likelihood of seeking PNC care in Ethiopia, Tanzania, and Benin [43–46]. This aligns with the findings of our study, which reported that women who didn't want more children were less likely to seek PNC care. Our study also found that educated women were less likely to seek PNC visits, which contrasts several studies [5,47,48], which report that more educated women (women with primary or secondary and above education) were more likely to seek PNC visits. The possible explanation for this could be that educated women were economically empowered to employ healthcare professionals for home services.

## Limitations and strengths

This study analysed pooled DHS datasets from 27 sub-Saharan African countries involving 58,648 women of reproductive age. Using nationally representative data provided a robust and comparative picture of the facilitators and barriers to maternity service utilisation across the region. The inclusion of a decade-long dataset enhances the reliability and generalisability of the findings, making an important contribution to advancing knowledge on maternal and newborn health coverage in SSA.

However, the exclusion of 19 countries due to unavailable data limits full regional representation. Reliance on self-reported data introduces the possibility of recall bias, while the cross-sectional design prevents determination of causality. In addition, the absence of a data reduction strategy, such as factor analysis, restricts the ability to assess the relative contributions of individual, household, and community-level factors. Finally, given the study's scope and eligible respondents, findings may not fully align with patterns reported in other literature.

## Implications for research and policy

Our findings have several important research and policy implications. A major opportunity for policy strengthening is a focus on improving health service accessibility by facilitating transportation and services for women living in rural areas, which would be a valuable strategy to improve maternal health service uptake. Education and employment were important facilitators for seeking 4+ANC visits and for health facility delivery in our study. Enhancing education is, therefore, an important strategy to help women achieve empowerment and employment and to help women and their partners make more informed healthcare decisions.

## Conclusion and recommendation

The study found that educated women of reproductive age in SSA were more likely to seek four or more ANC visits and deliver in healthcare facilities, while employed women were also more likely to access ANC, institutional delivery, and PNC within 48 hours of delivery. However, overall utilisation of maternity services in SSA is still very low, resulting in a lifetime

risk of maternal mortality. To address these challenges, we recommend targeted efforts to improve reproductive health education for women and girls, promoting informed health decision-making from an early age. This could include developing educational tools and workshops to encourage more frequent use of healthcare facilities for delivery, ANC, and PNC. Additionally, public health campaigns should leverage mass media, such as radio, to reach larger audiences in both rural and urban areas, helping to reduce preventable maternal and newborn health complications.

## Acknowledgments

The authors are grateful to MEASURE DHS for granting access to the datasets used in this study.

## Author contributions

**Conceptualization:** Julia Marie Hajjar, Obasanjo Bolarinwa.

**Data curation:** Obasanjo Bolarinwa, Oluwatobi Abel Alawode.

**Formal analysis:** Obasanjo Bolarinwa, Oluwatobi Abel Alawode.

**Funding acquisition:** Obasanjo Bolarinwa.

**Investigation:** Julia Marie Hajjar, Obasanjo Bolarinwa.

**Methodology:** Obasanjo Bolarinwa, Oluwatobi Abel Alawode.

**Project administration:** Obasanjo Bolarinwa.

**Resources:** Obasanjo Bolarinwa.

**Software:** Obasanjo Bolarinwa.

**Supervision:** Obasanjo Bolarinwa.

**Validation:** Obasanjo Bolarinwa.

**Visualization:** Obasanjo Bolarinwa.

**Writing – original draft:** Julia Marie Hajjar, Obasanjo Bolarinwa, Oluwatobi Abel Alawode, Adeolu Anthony Olagunju, Lawrence Jones-Esan, Sanni Yaya.

**Writing – review & editing:** Julia Marie Hajjar, Obasanjo Bolarinwa, Oluwatobi Abel Alawode, Adeolu Anthony Olagunju, Lawrence Jones-Esan, Sanni Yaya.

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
