## [Decision Letter · Decision Letter 0]

29 Aug 2024

Dear Dr. Afolabi Bolarinwa,

Thank you for submitting your manuscript to PLOS ONE. After careful consideration, we feel that it has merit but does not fully meet PLOS ONE’s publication criteria as it currently stands. Therefore, we invite you to submit a revised version of the manuscript that addresses the points raised during the review process.

We look forward to receiving your revised manuscript.

Kind regards,

Amanuel Yoseph, MPH

Academic Editor

PLOS ONE

Additional Editor Comments:

I critically reviewed your article entitled “Determinants of maternity services utilisation: a multi-level analysis across sub-Saharan Africa” which has the potential to add to the existing body of scientific knowledge, particularly in developing countries. However, there are some limitations in your article that need addressing before publication.

1. There are several grammatical and typological errors that authors need to carefully review.

2. Authors should extensively format manuscripts based on PLOS ONE journal style, including file naming. Avoid unnecessary italicizing and capitalization throughout the manuscript.

3. Make sure that your reference contains all the necessary details and PLOS ONE style.

Decision: Major revision

Reviewers' comments:

Reviewer's Responses to Questions

**Comments to the Author**

1. Is the manuscript technically sound, and do the data support the conclusions?

Reviewer #1: Yes

Reviewer #2: No

2. Has the statistical analysis been performed appropriately and rigorously?

Reviewer #1: Yes

Reviewer #2: Yes

3. Have the authors made all data underlying the findings in their manuscript fully available?

Reviewer #1: Yes

Reviewer #2: Yes

4. Is the manuscript presented in an intelligible fashion and written in standard English?

Reviewer #1: No

Reviewer #2: Yes

Reviewer #1: Thank you for your valuable work on this important topic. Your research addresses a prevalent and systematic issue, and your efforts are much appreciated. Here are my comments and suggestions for improving the manuscript:

Abstract Section

- Lines 29 and 30: "In 2020, approximately 800 women died daily as a result of largely preventable complications of pregnancy and delivery" – Please specify the location of these deaths. Is this a global statistic?

- Lines 55 and 56: "whilst women who were gainfully employed were more likely to utilize postnatal care within 48 hours of delivery" – The term "gainfully employed" is broad and should be clearly defined in the methods section.

Background Section

- Lines 63 and 64: This should be modified in line with the first comment in the abstract section.

- Lines 94 to 97: It would be more appropriate to include this information in the methodology section, specifically under outcome variables and explanatory variables.

- Overall, try to summarize and shorten the background section. Avoid repetitive ideas and combine related sentences for better clarity.

Results Section

- Lines 234 and 235: "For the household wealth index, 20% of the women reported being from poorer, middle, richer, and richest households" – This sentence needs restructuring to convey a clearer message.

- When reporting the results of the multilevel regression analysis, mention both comparison groups. For example, on page 17, lines 278 to 280: “it can be reported that the older a woman, the more likely she was to deliver at a health facility with women aged 25-34 and 35-49 being 19% (aOR=1.18; 95% CI:1.10 -1.26) and 39% (aOR=1.39; 95% CI: 1.28 – 1.51) significantly more likely to deliver at a health facility respectively” – More likely to deliver compared to which group? Ensure the comparison group is mentioned in all relevant instances.

Discussion Section

- The discussion should be rewritten, refined, and shortened. It should not merely repeat the results section. Instead, discuss possible reasons for the study's major findings and variations compared to other studies. Compare and contrast your results with the existing literature.

- For the result mentioned in lines 308 to 312, where women with primary and secondary education are more likely to have PNC compared to those with no education, discuss possible reasons for this finding.

- Discuss why women in polygamous marriages were less likely to have ANC visits but more likely to have PNC visits compared to those in monogamous relationships. Additionally, explore reasons why women with social media exposure, from households with higher wealth indices, and those in high literacy communities are less likely to have PNC visits.

Conclusion Section

- The conclusion should be rewritten and shortened. Focus on answering the objective of the study and providing clear, concise conclusions.

Your attention to these details will greatly enhance the clarity and impact of your manuscript. Thank you for considering these suggestions.

Reviewer #2: 1. Title:is better to be stated as Determinants of maternity services utilisation Across SubSaharan Africa

2. From line number 127 and 128, you stated as you considered 58648 reproductive women from 27 countries but not state how you pooled different country data.

3. Your propsed model for out come variable is good. but mony research that have been conducted on your research area using multilevel model. So is it apropriate model for your research problem? Why not using count models? ( needs your clear justification on it)

4. From line number 182; you you state “you have done model comparasion but you did not state for what purepose you do model comparasion.

5. From line number 251 you state “model 4” but from line number 182 you stated”model IV” , so the numberig method is better to be the same and the same is true for onthres.

6. “You stated as model comparasion have been done” but I did not get model compasion results for three models using AIC, BIC and other you stated method.

**Do you want your identity to be public for this peer review?** For information about this choice, including consent withdrawal, please see our Privacy Policy

Reviewer #1: No

Reviewer #2: No

---

## [Author Response · Author response to Decision Letter 1]

2 Oct 2024

Reviewers’ comments and replies

Reviewer #1: Thank you for your valuable work on this important topic. Your research addresses a prevalent and systematic issue, and your efforts are much appreciated. Here are my comments and suggestions for improving the manuscript:

Reply

Thank you for taking the time to review our manuscript and we thank you for your review and comments that have improved our work.

Abstract Section

- Lines 29 and 30: "In 2020, approximately 800 women died daily as a result of largely preventable complications of pregnancy and delivery" – Please specify the location of these deaths. Is this a global statistic?

Reply

Yes, it’s a global statistic, and this has been added. Line 30 and 63.

- Lines 55 and 56: "whilst women who were gainfully employed were more likely to utilise postnatal care within 48 hours of delivery" – The term "gainfully employed" is broad and should be clearly defined in the methods section.

Reply

The word “gainfully employed” has been changed to “employed” throughout the manuscript; information about employment was explained under the methods section in line 150.

Background Section

- Lines 63 and 64: This should be modified in line with the first comment in the abstract section.

Reply

Yes, it’s a global statistic, and this has been added. Line 30 and 63.

- Lines 94 to 97: It would be more appropriate to include this information in the methodology section, specifically under outcome variables and explanatory variables.

Reply

Yes, we included it. Lines 135 to 142.

- Overall, try to summarise and shorten the background section. Avoid repetitive ideas and combine related sentences for better clarity.

Reply

The background section has been restructured to avoid repetitions of ideas.

Results Section

- Lines 234 and 235: "For the household wealth index, 20% of the women reported being from poorer, middle, richer, and richest households" – This sentence needs restructuring to convey a clearer message.

Reply

The paragraph has been rewritten to convey a clear message. Page 10, lines 234 to 237.

- When reporting the results of the multilevel regression analysis, mention both comparison groups. For example, on page 17, lines 278 to 280: “it can be reported that the older a woman, the more likely she was to deliver at a health facility with women aged 25-34 and 35-49 being 19% (aOR=1.18; 95% CI:1.10 -1.26) and 39% (aOR=1.39; 95% CI: 1.28 – 1.51) significantly more likely to deliver at a health facility respectively” – More likely to deliver compared to which group? Ensure the comparison group is mentioned in all relevant instances.

Reply

Thank you for the observation. This has been corrected throughout the manuscript.

Discussion Section

- The discussion should be rewritten, refined, and shortened. It should not merely repeat the results section. Instead, discuss possible reasons for the study's major findings and variations compared to other studies. Compare and contrast your results with the existing literature.

Reply

This section has been rewritten and reduced.

- For the result mentioned in lines 308 to 312, where women with primary and secondary education are more likely to have PNC compared to those with no education, discuss possible reasons for this finding.

Reply

More information about this has been provided in the discussion section in line with the study results.

- Discuss why women in polygamous marriages were less likely to have ANC visits but more likely to have PNC visits compared to those in monogamous relationships. Additionally, explore reasons why women with social media exposure, from households with higher wealth indices, and those in high literacy communities are less likely to have PNC visits.

Reply

More information about this has been provided in the discussion section in line with the study results.

Conclusion Section

- The conclusion should be rewritten and shortened. Focus on answering the objective of the study and providing clear, concise conclusions.

Reply

This section has been rewritten and reduced.

Your attention to these details will greatly enhance the clarity and impact of your manuscript. Thank you for considering these suggestions.

Reply

We have checked through the manuscript again and have made substantial changes to improve the manuscript.

Reviewer #2: Comment

General comments:

--writeup in introduction is well

- Writeup in method is a litle bit good.

-result write up have good floow and detail

-the overall result will have significant impact

Reply

Thank you for taking the time to review our manuscript and we thank you for your review and comments that have improved our work.

1. Title:is better to be stated as Determinants of maternity services utilisation Across SubSaharan Africa

Reply

The title has been changed to reflect this.

2. From line number 127 and 128, you stated as you considered 58648 reproductive women from 27 countries but not state how you pooled different country data.

Reply

Information about this has been added to page 6, lines 173 to 175, under statistical analyses.

3. Your propsed model for out come variable is good. but mony research that have been conducted on your research area using multilevel model. So is it apropriate model for your research problem? Why not using count models? ( needs your clear justification on it)

Reply

Thank you for calling our attention to this. Yes, we believed this is the appropriate model because of the nature of the dataset, and because we considered three outcome variables in this study, we believed it’s appropriate to consider using the same model for the three outcomes given the fact that “facility delivery” is dichotomous.

4. From line number 182; you you state “you have done model comparasion but you did not state for what purepose you do model comparasion.

Reply

This information has been added to ensure better understanding “to show which model significantly improves over others whilst ensuring the appropriate model with goodness of fit is selected.”

5. From line number 251 you state “model 4” but from line number 182 you stated”model IV” , so the numberig method is better to be the same and the same is true for onthres.

Reply

This has been changed to “model I to model IV” in the same format.

6. “You stated as model comparasion have been done” but I did not get model compasion results for three models using AIC, BIC and other you stated method.

Reply

The results were stated in the tables except for BIC. All other results have been reported in the results section, and the BIC claim has been deleted.

---

## [Decision Letter · Decision Letter 1]

9 Apr 2025

Dear Dr. Bolarinwa,

Thank you for submitting your manuscript to PLOS ONE. After careful consideration, we feel that it has merit but does not fully meet PLOS ONE’s publication criteria as it currently stands. Therefore, we invite you to submit a revised version of the manuscript that addresses the points raised during the review process.

The manuscript has been evaluated by two reviewers, and their comments are available below. Although reviewer 1 is satisfied with the revised manuscript, reviewer 3 has raised a concern that needs attention.



We look forward to receiving your revised manuscript.

Kind regards,

Steve Zimmerman, PhD

Senior Editor, PLOS One

Journal Requirements:

Reviewers' comments:

Reviewer's Responses to Questions

**Comments to the Author**

Reviewer #1: All comments have been addressed

Reviewer #3: (No Response)

2. Is the manuscript technically sound, and do the data support the conclusions?

Reviewer #1: Yes

Reviewer #3: (No Response)

3. Has the statistical analysis been performed appropriately and rigorously?

Reviewer #1: Yes

Reviewer #3: (No Response)

4. Have the authors made all data underlying the findings in their manuscript fully available?

Reviewer #1: Yes

Reviewer #3: (No Response)

5. Is the manuscript presented in an intelligible fashion and written in standard English?

Reviewer #1: Yes

Reviewer #3: (No Response)

Reviewer #1: (No Response)

Reviewer #3: This retrospective cross-sectional study employed Demographic Health Surveys (DHS) conducted in 27 countries in SSA. The outcome and explanatory variables were adequately defined.

On Line 164 the authors state , “…. three-level multi-level logistic regression….” Do they actually mean, “three-level multiple logistic regression…”?

Tables 3,4 and 5 are well constructed as one would expect a hierarchical presentation. Since this was a multilevel look, the authors did put a descriptive paragraph at the end of the Tables 3,4, and 5 presentation stating the relative merits of the three constructs in some way. However, this could have been a bit more sophisticated beyond the model fit statistics with a Factor analysis type structure if individual, household and community separate out to three meaningful constructs to determine which grouping is the most statistically significant contributor to the three endpoints being investigated. Why , no attempt for a Data reduction strategy in view of the many variables being examined separately? There is no explanation in the discussion section as to the statistical overall comparison of the three levels (constructs) as to their relative importance individually or why that was not considered. This could possibly be a study limitation from the analytical perspective.

**Do you want your identity to be public for this peer review?** For information about this choice, including consent withdrawal, please see our Privacy Policy

Reviewer #1: No

Reviewer #3: No

---

## [Author Response · Author response to Decision Letter 2]

25 Aug 2025

Reviewers comments and replies

Reviewer #1: (No Response)

Reply

Thank you for taking the time to review our manuscript. We greatly appreciate your feedback and constructive comments.

Reviewer #3: This retrospective cross-sectional study employed Demographic Health Surveys (DHS) conducted in 27 countries in SSA. The outcome and explanatory variables were adequately defined.

Reply

Thank you for taking the time to review our manuscript. We greatly appreciate your feedback and constructive comments.

On Line 164 the authors state , “…. three-level multi-level logistic regression….” Do they actually mean, “three-level multiple logistic regression…”?

Reply

This study employed a three-level multilevel logistic regression analysis rather than a multiple regression approach, as it accounted for unexplained variations across all levels. Models 0 to IV were assessed using log-likelihood ratio and Akaike Information Criterion (AIC), as presented in Tables 3 to 5.

Tables 3,4 and 5 are well constructed as one would expect a hierarchical presentation. Since this was a multilevel look, the authors did put a descriptive paragraph at the end of the Tables 3,4, and 5 presentation stating the relative merits of the three constructs in some way. However, this could have been a bit more sophisticated beyond the model fit statistics with a Factor analysis type structure if individual, household and community separate out to three meaningful constructs to determine which grouping is the most statistically significant contributor to the three endpoints being investigated. Why , no attempt for a Data reduction strategy in view of the many variables being examined separately? There is no explanation in the discussion section as to the statistical overall comparison of the three levels (constructs) as to their relative importance individually or why that was not considered. This could possibly be a study limitation from the analytical perspective.

Reply

Thank you for raising this important observation. We have addressed it in the limitations section of the revised manuscript. The updated Strengths and Limitations section now reads as follows:

“This study analysed pooled DHS datasets from 27 sub-Saharan African countries involving 58,648 women of reproductive age. Using nationally representative data provided a robust and comparative picture of the facilitators and barriers to maternity service utilisation across the region. The inclusion of a decade-long dataset enhances the reliability and generalisability of the findings, making an important contribution to advancing knowledge on maternal and newborn health coverage in SSA. However, the exclusion of 19 countries due to unavailable data limits full regional representation. Reliance on self-reported data introduces the possibility of recall bias, while the cross-sectional design prevents determination of causality. In addition, the absence of a data reduction strategy, such as factor analysis, restricts the ability to assess the relative contributions of individual, household, and community-level factors. Finally, given the study’s scope and eligible respondents, findings may not fully align with patterns reported in other literature.”

---

## [Decision Letter · Decision Letter 2]

8 Jan 2026

Determinants of maternity services utilisation among women of reproductive age across sub-Saharan Africa

PONE-D-24-26208R2

Dear Dr. Bolarinwa,

We’re pleased to inform you that your manuscript has been judged scientifically suitable for publication and will be formally accepted for publication once it meets all outstanding technical requirements.

Kind regards,

Mubarick Nungbaso Asumah

Academic Editor

PLOS One

Additional Editor Comments (optional):

Reviewer #3:

Reviewers' comments:

Reviewer's Responses to Questions

**Comments to the Author**

Reviewer #3: All comments have been addressed

2. Is the manuscript technically sound, and do the data support the conclusions?

Reviewer #3: (No Response)

3. Has the statistical analysis been performed appropriately and rigorously?

Reviewer #3: (No Response)

4. Have the authors made all data underlying the findings in their manuscript fully available?

Reviewer #3: (No Response)

5. Is the manuscript presented in an intelligible fashion and written in standard English?

Reviewer #3: (No Response)

Reviewer #3: (No Response)

**Do you want your identity to be public for this peer review?** For information about this choice, including consent withdrawal, please see our Privacy Policy

Reviewer #3: No

---

## [Editor Report · Acceptance letter]

PONE-D-24-26208R2

PLOS One

Dear Dr. Bolarinwa,

I'm pleased to inform you that your manuscript has been deemed suitable for publication in PLOS One. Congratulations! Your manuscript is now being handed over to our production team.

Kind regards,

on behalf of

Mr. Mubarick Nungbaso Asumah

Academic Editor

PLOS One